**Subject Category:**
Biology (whole organism)

health and disease and epidemiology

*Phlebotomus*, GIS-overlay, MAXENT, ecological niche modelling, neglected tropical diseases, sandflies

**Author for correspondence:**
Sarah Cunze
e-mail: sarahcunze@gmail.com

# Leishmaniasis in Eurasia and Africa: geographical distribution of vector species and pathogens

Sarah Cunze[1,2], Judith Kochmann[1,2], Lisa K. Koch[1,2], Korbinian J. Q. Hasselmann[1,2] and Sven Klimpel[1,2]

[1]Institute of Ecology, Evolution and Diversity, Goethe-University, Max-von-Laue-Strasse 13, 60438 Frankfurt, M., Germany
[2]Senckenberg Biodiversity and Climate Research Centre, Senckenberganlage 25, 60325 Frankfurt, M., Germany

SC, 0000-0003-3319-2590

Leishmaniasis is a vector-borne disease with a broad global occurrence and an increasing number of recorded cases; however, it is still one of the world's most neglected diseases. We here provide climatic suitability maps generated by means of an ecological niche modelling approach for 32 *Phlebotomus* vector species with proven or suspected vector competence for five *Leishmania* pathogens occurring in Eurasia and Africa. A GIS-based spatial overlay analysis was then used to compare the distributional patterns of vectors and pathogens to help evaluate the vector species–pathogen relationship currently found in the literature. Based on this single factor of vector incrimination, that is, co-occurrence of both vector and pathogen, most of the pathogens occurred with at least one of the associated vector species. In the case of *L. donovani*, only a not yet confirmed vector species, *P. rodhaini*, could explain the occurrence of the pathogen in regions of Africa. *Phlebotomus alexandri* and *P. longiductus* on the other hand, proven vector species of *L. donovani*, do not seem to qualify as vectors for the pathogen. Their distribution is restricted to northern latitudes and does not match the pathogen's distribution, which lies in southern latitudes. Other more locally confined mismatches were discussed for each pathogen species. The comparative geographical GIS-overlay of vector species and pathogens functions as a first indication that testing and re-evaluation of some pathogen–vector relationships might be worthwhile to improve risk assessments of leishmaniasis.

## 1. Background

Leishmaniasis is a vector-borne disease with a broad global distribution and an increasing number of recorded cases

worldwide [1,2]. However, it is still one of the world's most neglected diseases [3]. Over the last decades, the disease has been found to expand geographically with a global increase of cases of visceral and cutaneous leishmaniasis [4] increasing the public health problems associated with the disease epidemics. The reported range expansion of the diseases has been associated with range expansions of vector populations in response to climate change [5]. Leishmaniasis is caused by protozoan parasites of the genus *Leishmania*. The transmission can either be zoonotic and/or anthroponotic through the bite of an infected female phlebotomine sandfly [6]. In Eurasia and Africa, all vector-competent sandfly species belong to the genus *Phlebotomus* [7]. Cutaneous leishmaniasis (CL) is the most common form of leishmaniasis. In the 'old world', it is caused by five currently recognized *Leishmania* species: *L. major*, *L. tropica* and *L. aethiopica* (being main causative parasites) as well as *L. infantum* and *L. donovani*. Visceral leishmaniasis (VL), another common and more severe form of leishmaniasis, is only associated with the *Leishmania* species *L. infantum* and *L. donovani* [8–11]. The specific *Leishmania* species cause different clinical symptoms in humans [12,13].

A successful transmission requires the presence of pathogen, vector and host species [14]. Thus, when estimating disease risk the occurrence of the vector species in addition to the distributions of the disease-causing pathogen and the host species are of major importance [14,15]. For a better understanding of the epidemiology of vector-borne diseases, comparisons between the geographical distributions of vectors, pathogens and disease cases using geographical information systems (GIS) have been suggested as one of the vector incrimination criteria adopted by WHO [3].

Another important component required for the completion of the *Leishmania* life cycle is the reservoir host [15]. Generally, the pathogen–vector–reservoir host relations are rather complex and have been shown to vary regionally and temporally. Several species, from small to large, wild, domestic and synanthropic mammals have been reported as reservoir hosts for *Leishmania* spp., among them rock hyraxes (especially for the Indian subcontinent with India, Nepal and Bangladesh, and east Africa with Ethiopia and Kenya), rodents, foxes, dogs, cats and other domestic animals [6]. Among the five *Leishmania* species considered here, *L. donovani* and *L. tropica* are assumed to be anthroponotic with humans as reservoir hosts, having human-to-human transmission through the vector, but there is some evidence for the possible involvement of zoonotic transmission as well [6]. Although the presence or the absence of reservoirs is crucial for the distribution of leishmaniasis, reservoir hosts were not included in the analysis. One of the reasons is their much broader range of distribution [15,16].

Here, we focus on leishmaniasis in Europe, Northern and Central Africa, and parts of Asia (73° N–10° S, 19° W–94° E), accounting for the distributional patterns of the five *Leishmania* parasite species and their associated vector species (*Phlebotomus* spp.) as well as their interrelationship. Most of these vector species are considered to be pathogen-specific (i.e. each sandfly species typically transmits only one *Leishmania* species [6]), which should be reflected in the distribution patterns of the respective pathogens and vectors (see [15], which examined this pattern for Libya). Mismatches between the assumed patterns may help to evaluate the recognized vector species–pathogen relationship. We assume that wherever a pathogen is known to occur, at least one of the associated vector species must also be present. In the case of mismatches, the following scenarios are possible for a specific location: (i) The occurrence of pathogens without competent/suspected vectors may point to a species with a currently unknown vector potential, (ii) the absence of pathogens with the presence of competent/suspected vectors might point towards different temperature requirements of the pathogen in comparison to the sandfly vector species, limited dispersal abilities or biogeographic barriers for infected vectors, or an incorrect assumption about the vector potential. On the other hand, areas modelled to be suitable but currently unrecognized as distributional areas for a certain vector species, could potentially be suitable regions, which are not inhabited due to biogeographic constraints, dispersal limitations, or not yet recognized due to the lack of entomological surveillance.

We focused on the comparison between the geographical distributions of *Phlebotomus* vector species and the distributions of pathogens. For the distribution of the vector species, we used an ecological niche modelling approach to establish climatic suitability maps, whereas for the distribution of pathogens we referred to pre-existing polygon data.

# 2. Material and methods

## 2.1. Vector–pathogen relationship

As a first step, relevant information about the relationship between *Phlebotomus* vectors and *Leishmania* pathogens in Eurasia and Africa were gathered from the literature. Each vector species was

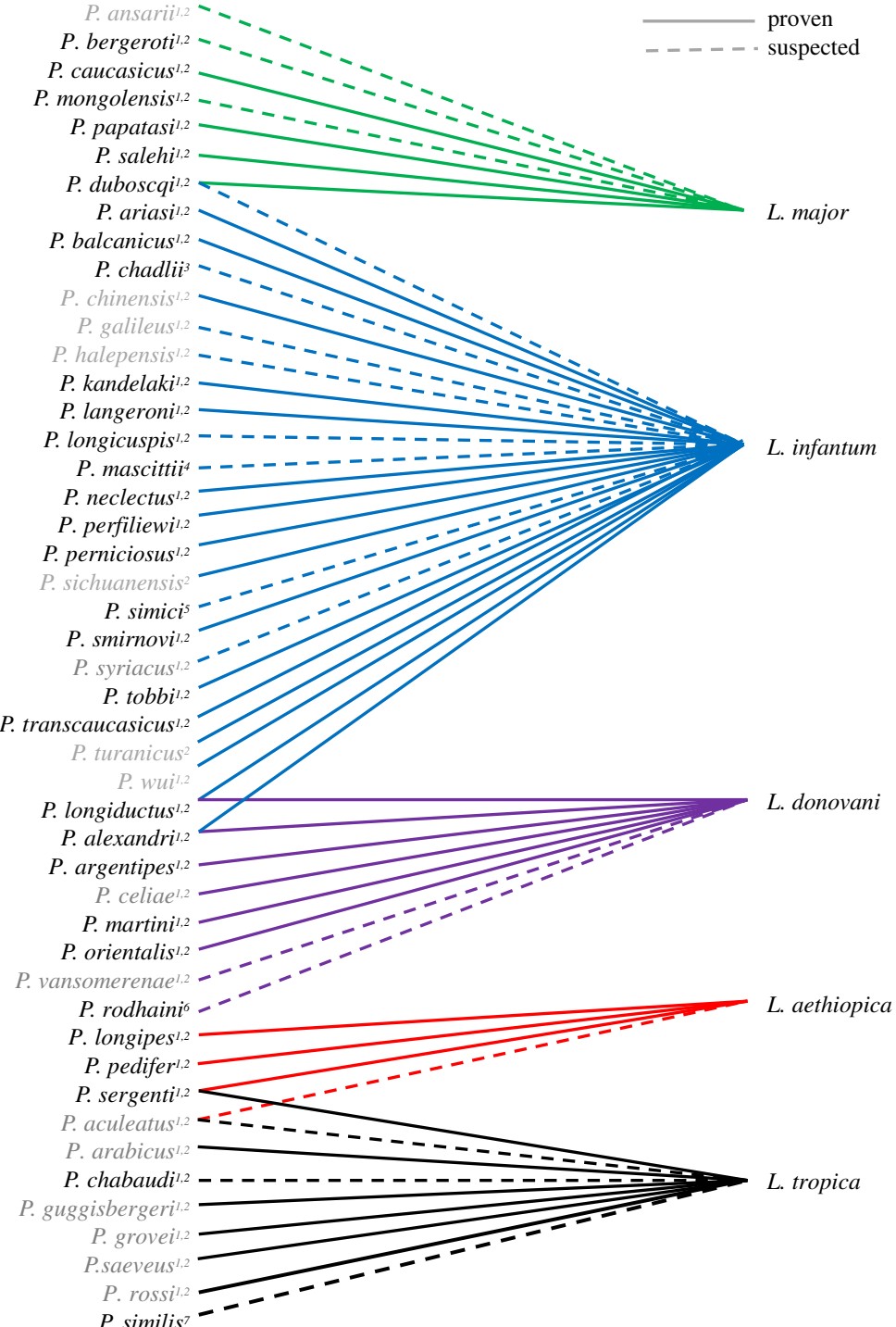

**Figure 1.** Vector–pathogen relationship for large parts of Eurasia and Africa. Shown are 47 vector species (*Phlebotomus* spp.) and five pathogens (*Leishmania* spp.). Vector-competence of the *Phlebotomus* species was confirmed (solid lines) or not yet confirmed but strongly suggested (dotted lines) according to 1: WHO [3], 2: Maroli *et al.* [10], 3: Berdjane-Brouk *et al.* [17], 4: Obwaller *et al.* [18], 5: Naucke [19], 6: Elnaiem *et al.* [20], 7: Depaquit *et al.* [21]. Phlebotomus names in grey (*N* = 15) indicate that the information on the geographical distribution was insufficient for ecological niche modelling and further comparative analysis.

linked to the pathogens it is able to transmit (figure 1). As the vector competence of some sandfly species has yet to be confirmed, species were separated into two groups ('confirmed' and 'suspected' vector species).

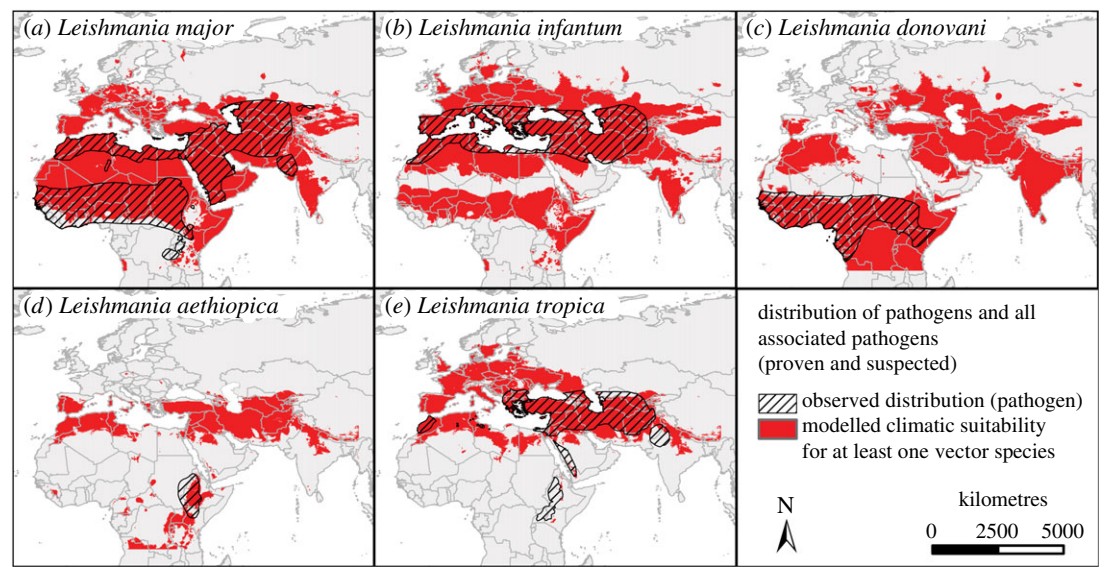

**Figure 2.** Climatic suitability of vector species and distribution of associated pathogens. Hatched areas indicate the observed distribution of the five *Leishmania* species (according to [3,16,22,23]), red indicates areas with suitable climatic conditions based on the binary modelling results using equal training sensitivity and specificity threshold projected for at least one of the associated vector species (with confirmed or strongly suspected vector-competence for the respective *Leishmania* species). See the electronic supplementary material, figure S1 for additional information on individual vector species occurrences.

## 2.2. Occurrence data

Distribution maps of the *Leishmania* pathogens were available from different sources [3,22–24]. The information of these maps (polygon data) was combined resulting in five *Leishmania* species-specific distribution maps (figure 2).

Occurrence point data, and no polygon data, was available for the different *Phlebotomus* species. In order to find all potentially suited habitats for the vector species and to not solely rely on the generally low number of recorded presences, we used an ecological niche modelling (ENM) approach. The occurrence data for ecological niche modelling of the *Phlebotomus* species was mainly obtained from [25] and supplemented by our own literature research (e.g. [26–36]). Electronic supplementary material, table S3 lists the occurrence records of *Phlebotomus* species used for ecological niche modelling.

## 2.3. Ecological niche modelling and environmental data

We used the maximum entropy modelling approach implemented in the software MAXENT ([37], v. 3.3.3 k). The ecological niche model was based on eight climatic variables provided by WORLDCLIM (v. 1.4, [38]), downloaded with a spatial resolution of 5 arc minutes (this corresponds to spatial resolution of about 10 km$^2$ at the equator). This resolution is in accordance with the resolution of the occurrence records. Data represent averages for 1960–1990. Four temperature variables and four precipitation variables, reflecting the mean, minima, maxima and variability of annual patterns, were considered. More specifically, annual mean temperature (bio01), temperature seasonality, i.e. the standard deviation of monthly temperature (bio04), maximal temperature of the warmest month (bio05), minimal temperature of the coldest month (bio06), annual precipitation (bio12), precipitation seasonality (bio15), precipitation of the wettest month (bio13) and precipitation of the driest month (bio14) were chosen as predictive variables. These variables were considered to be ecologically relevant for the sandfly species [39–41]. Some of these variables are highly intercorrelated, but in contrast to other ENM approaches MAXENT deals well with intercorrelation, so there is less need to remove correlated variables [42]. Other factors such as land cover, elevation, vegetation indices or soil parameters also affect the distribution patterns of sandfly species. But in this study we focus on climatic suitability.

The following modifications of the MAXENT default settings were made: use of only linear, quadratic and product features (cf. [43]), maximum iteration number of 500 000 and convergence threshold of 0.00000001. The permutation importance values were taken as an estimator for the relative contribution of the different variables to the MAXENT model.

To allow for a comparison of the polygonal distribution areas of the *Leishmania* pathogens and the vectors, binary climatic suitability maps were created, using equal training sensitivity and specificity threshold [44], whose good performance was shown in comparison to other criteria, e.g. by [45]. This threshold minimizes the false positive and negative rates and, similarly, to maximizes the true negative and positive rates, placing equal weight on presences and absences [46].

## 2.4. GIS-overlay of vector and pathogen distributions

In order to assess the common distribution areas of pathogens and associated vectors, areas were overlaid. The areas with at least one of the associated vector species modelled to find suitable climatic conditions (based on the binary modelling results using equal training sensitivity and specificity threshold) were displayed in red and the areas with the associated *Leishmania* pathogen were displayed as hatched areas (figure 2). Maps for each single vector species, depicting occurrence records and modelled climatic suitability, were also created (electronic supplementary material, figure S1).

## 2.5. One-variable response curves

The distributional patterns of vector species indicate different climatic requirements, which are usually presented in one-variable response curves. These curves reflect the modelled niche functions (i.e. species–environment relationship) considering only one specific variable and one species. The occurrence points for the pathogens were derived from overlapping the disease occurrences provided by Pigott *et al.* [1] with the pathogen distribution polygons according to the earlier studies [3,22–24].

The one-variable response curves were exemplarily displayed for *L. donovani* and its associated vector species, considering annual mean temperature.

## 2.6. Software

We used the maximum entropy modelling approach implemented in the software MAXENT [37] (v. 3.3.3 k) for modelling climatic suitability. Further analyses were performed in R [47] and ESRI ArcGIS [48] (Release 10.3). Maps were built using ESRI ArcGIS [48] (Release 10.3).

# 3. Results

## 3.1. Vector–pathogen–disease relationship

We accounted for 48 *Phlebotomus* species occurring in Africa, Europe and Asia with proven or assumed vector-competence [3,10,17–21,49,50]. The associations between these 48 vector species (*Phlebotomus* spp.), and the five pathogen species (*Leishmania* spp.) are summarized in figure 1. For some of the *Phlebotomus* species, the present core distribution lies outside the considered study area, e.g. *P. chinensis* and *P. wui* with a core distribution in China. For other species, only poor distributional information is currently available (rare species or endemic species with a small geographical range, e.g. *P. celiae*). As a result, 16 out of 48 *Phlebotomus* species were not taken into account for further analyses due to the limited number of available occurrence records within the study area.

Most of the considered vector species are considered pathogen-specific, i.e. they are vector-competent for only one *Leishmania* species [10]. However, according to [10] and the latest WHO report [3], a few exceptions exist: *P. alexandri* has been suggested to transmit *L. donovani* and *L. infantum* in parts of China and other countries; *P. sergenti* is considered a transmitter of *L. tropica* and *L. aethiopica* in parts of Ethiopia; *P. duboscqi* is considered vector-competent for *L. major* and might additionally be a competent vector for *L. infantum* in Gambia; *P. longiductus* is suspected to be vector-competent for *L. infantum* and for *L. donovani* in China. *P. aculeatus* is suspected to transmit *L. aethiopica* in Kenya according to [3] or *L. tropica* according to [10].

## 3.2. Modelled climatic suitability for the considered *Phlebotomus* species

The models of climatic suitability for all considered 32 *Phlebotomus* species reflect the observed occurrence distributions well (electronic supplementary material, figure S1). For most species, the modelled area reaches beyond the area with observed occurrences (e.g. *P. argentipes* or *P. simici*; electronic supplementary material, figure S1).

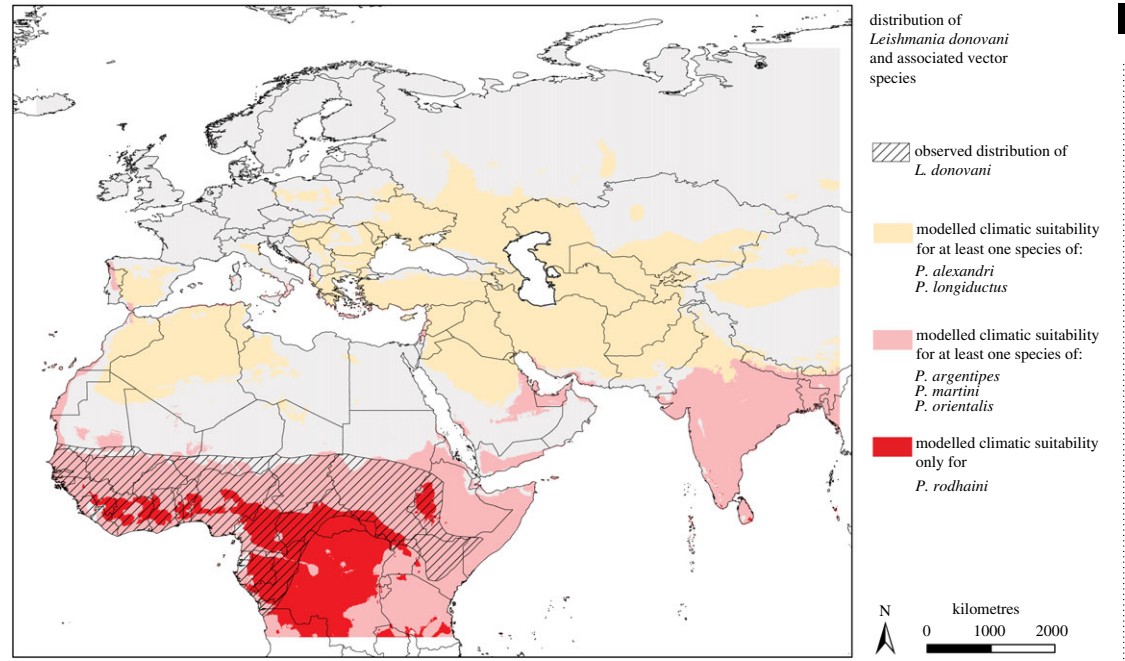

**Figure 3.** Geographical distribution of *Leishmania donovani* and its associated vector species. Hatched areas indicate the observed distribution of *L. donovani*. Coloured areas indicate modelled suitability for at least one vector species of *L. donovani*: orange, northern group (*P. alexandri*, *P. longiductus*); light red, southern group (*P. argentipes*, *P. martini*, *P. orientalis* and *P. rodhaini*); dark red, *P. rodhaini* and none of the other associated vector species.

Considering the average rank over all considered vector species, temperature seasonality (bio04) and minimal temperature of the coldest month (bio06) were identified as the variables best explaining the distributional patterns of *Phlebotomus* species, followed by annual mean temperature (bio01). The temperature variables generally showed higher contributions to the MAXENT models than the precipitation variables, with some exceptions. In the case of *P. chadii*, *P. langeroni*, *P. mongolensis*, *P. similis* and *P. transcaucasicus* precipitation of the driest month (bio14) was modelled to be the first or second most important predictor variable. The permutation importance values are summarized in electronic supplementary material, table S2.

## 3.3. Distribution of *Leishmania* pathogens and their vector species

The distribution patterns of the five pathogen species are largely in accordance with those of the associated vector species (figure 2). The observed distribution of *L. major*, *L. infantum* and *L. donovani* overlaps greatly with areas projected to be climatically suitable for at least one of the associated vector species (except small areas at the edges of the pathogen distribution for *L. major* (figure 2*a*) and *Leishmania donovani* (figure 2*c*)). However, there are quite a few regions that have been modelled as climatically suitable for vector species without occurrence of the pathogen. *Leishmania aethiopica* is only known from Ethiopia and adjacent areas in Eritrea, Sudan and Kenya. Only parts of these regions are modelled to provide climatically suitable conditions for at least one of the associated vector species (figure 2*d*). Similarly, *L. tropica* matches the distributions of the associated vectors only in some areas (Eurasia and Morocco); in parts of Ethiopia, Kenya and Uganda, and in Saudi Arabia, India and Pakistan the pathogen is assumed to occur but none of the associated vector species seem to find climatically suited habitats there (figure 2*e*). Species-specific vector distribution patterns and observed occurrence points provide further information (electronic supplementary material, figure S1).

## 3.4. *Leishmania donovani* and its associated vector species

We specifically focused on the distributional patterns of *L. donovani* and its associated vector species. The associated vector species of *L. donovani* might be classified into two groups (figure 3) according to the patterns of geographical distributions. The one group, referred to as 'southern group', comprising

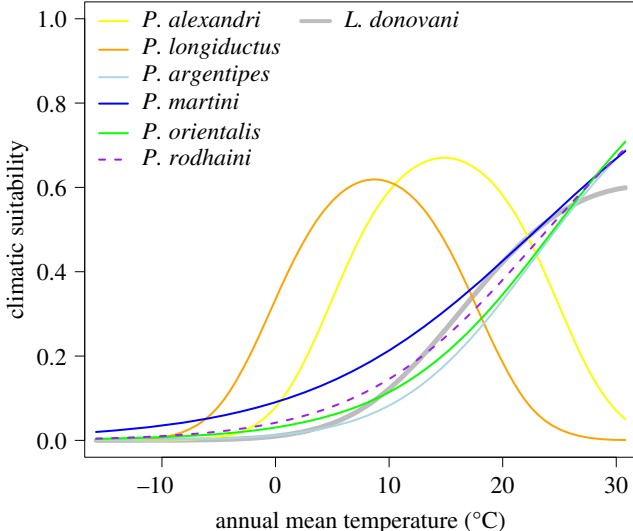

**Figure 4.** One-variable response curves (niche functions) for *L. donovani* and associated vector species. Modelled climatic suitability depending on the annual mean temperature (°C) for the vector species (proven vector-competence, solid line; suspected vector-competence, dotted line) and the associated *Leishmania* pathogen (bold line).

*P. argentipes*, *P. martini*, *P. orientalis* as well as *P. rodhaini* (depicted in light red and dark red in figure 3), is mainly distributed in tropical areas and overlaps with the distribution of *L. donovani* (hatched area in figure 3). The other group, referred to as 'northern group', comprises *P. alexandri* and *P. longiductus* (depicted in orange in figure 3); this group does not overlap with the distribution of the pathogen within the study area. Within the southern group, we additionally distinguished between vector species with an established vector-competence (these are: *P. argentipes*, *P. martini* and *P. orientalis)*, and *P. rodhaini* for which vector-competence has only been assumed [20]. It can be noted that the area modelled to be climatically suitable for *P. rodhaini* goes beyond the area of the other species of the southern group (figure 3).

The aforementioned differences between the two groups of vector species associated with *L. donovani* in the distributional patterns in the geographical space are also reflected by differences in the modelled temperature niche functions (figure 4). The modelled optima of annual mean temperature (bio01) is lower for the northern group of vector species (*P. alexandri* and *P. longiductus*), whereas curves for species found in more tropical climates with monotonously increasing temperature niche curves are more similar to the optimum curve of the pathogen *L. donovani* (figure 4). Overall, it can be noted that the southern group of vector species is more closely connected to the pathogen *L. donovani* regarding its geographical distribution as well as the modelled temperature niche than the northern group of vector species.

# 4. Discussion

## 4.1. Modelled climatic suitability for vector species of leishmaniasis

Reliable data of species' distribution is often difficult to obtain, but is of great importance for estimating disease risks. Due to the lack of reliable distribution maps for the large number of vector species, an ecological niche modelling approach was used to generate climatic suitability maps revealing potential areas of vector distribution. Principal validity of the application of ecological niche modelling based on climatic variables for *Phlebotomus* species has been previously shown in [15,51–57].

As patterns of modelled climatic suitability reflected the observed occurrences of the 32 *Phlebotomus* species well, with area under the receiver operating curve (AUC) values of above 0.88 for all species, the modelling results could be used as estimators for the distributional patterns of the considered vector species. For some *Phlebotomus* species, the area with modelled climatic suitability goes beyond the area with observed occurrence. Reasons for this could be the following: (i) a sampling bias for some of the species in the occurrence data, i.e. the species have been recorded only in some parts of its distributional range; (ii) insufficient time or dispersal barriers like oceans or mountain ranges might have prevented species to reach all places with suitable climatic conditions (so far); (iii) disregarding

one or more factors others than climate being important drivers for the distributional patterns may yield an overestimation of the potential distribution.

## 4.2. Vector–pathogen relationship

In general, the distributional patterns of the *Leishmania* pathogens are in good accordance with the areas modelled to be climatically suitable for at least one of the associated vector species. However, small areas at the edges of a pathogen distribution seemingly lack suitable habitats for the associated vectors. This is most obvious for the southern edge of *L. major, L. aethiopica* and *L. tropica*. In the case of *L. major*, the vector distribution (proven and suspected vectors) does not entirely cover the distribution of the pathogen. The southern edge of the *L. major* distribution lies in the vegetation zone covered by tropical rainforest, and may be the habitat of a currently unknown vector species for this pathogen. Another reason might be an underestimation of the climatic suitability for a vector species, which was lost due to dichotomy usage. Furthermore, *P. ansarii*, suspected to be vector competent for *L. major*, but not considered here due to the lack of occurrence records, has only been reported for Iran, so far [58]. In the case of *L. aethiopica*, the currently known distribution is relatively small in comparison to the other *Leishmania* species. Records of *L. aethiopica* are currently restricted to the highlands of Ethiopia and Kenya [59]. The aforementioned mismatch can probably be ascribed to the fact that some of the associated vector species were not included in the analysis due to insufficient data, e.g. *P. aculeatus* which is assumed to be vector-competent for *L. aethiopica* and known to occur in Kenya [60]. However, the vector competence of *P. aculeatus* has been questioned by Krayter *et al.* [59], referring to the morphological indistinguishability between *P. aculeatus* and *P. pedifer*. *Phlebotomus pedifer,* as well as *P. longipes,* are the two main vectors for *L. aethiopica* [59], but their models of climatic suitability were based on a very small number of occurrence points (6 for *P. pedifer*, 14 for *P. longipes*) Thus, the distribution of these species might be underrepresented in these data, which may have led to an underestimation of the area with modelled climatically suitability for these species.

The distribution of *L. tropica* is largely covered by area modelled to be suitable for *P. sergenti* (except the area around Kenya). Besides *P. sergenti*, the two vector species with suspected vector competence, *P. chabaudii* and *P. simici*, were considered in the GIS-overlay in figure 2*e*. The discrepancies in parts of Ethiopia, Kenya and Uganda, and in Saudi Arabia, India and Pakistan are likely to be explained by the fact that a number of known vector species (e.g. *P. arabicus*, *P. guggisbergeri*, *P. groveri* and *P. rossi*) were not considered in the modelling due to lack of occurrence data. Especially for a sandfly as a species of small body size, a potential sampling bias due to an underestimation of occurrences in certain areas is quite likely.

For all species, the area modelled to be climatically suitable for the vector species reaches beyond the observed pathogen distribution area. The most plausible explanation might not generally be an erroneous vector–pathogen association, but rather that the vector species is able to occur under a broader temperature range compared to the pathogen, which may require higher temperatures for its development and transmission [61]. Another explanation might be that infected vectors did not reach some parts of the areas with potentially suitable climatic conditions yet, e.g. due to biogeographic dispersal barriers or their weak flying abilities. For example, in the case of *L. major*, the observed pathogen distribution area covers parts of the Sahara desert with extreme climatic conditions, hostile for many species and requiring species to be adapted. Among the seven associated vector species of *L. major*, *P. bergeroti* was the only one with modelled climatic suitability in the central parts of the Sahara.

We had a closer look at the distributional patterns of *L. donovani* and its associated vector species, which can be distinguished into two groups according to the vectors' distribution pattern: one group occurring further north in the study area (comprising *P. alexandri* and *P. longiductus*) and a second group occurring further south in the study area (comprising the species *P. argentipes*, *P. martini*, *P. orientalis* as well as *P. rodhaini*). This separation of the associated vector species into two groups can also be found regarding the one-variable response curve with the annual mean temperature as explanatory variable. Species of the first group show optima under lower temperatures, whereas the niches of the second group, including *L. donovani*, show optima at warmer temperatures. The vector species of the 'southern' group are modelled to co-occur with *L. donovani*, whereas *P. alexandri* and *P. longiductus* of the 'northern' group do not show any overlap with the observed pathogen distribution.

The area modelled to be climatically suitable for the species of the second southern group with *P. argentipes*, *P. martini*, *P. orientalis* and *P. rodhaini* is in good accordance with the observed distribution of the pathogen, i.e. the area modelled to be climatically suitable for at least one of the associated vector species covers and extends beyond the known pathogen distribution. However, distinguishing between proven and suspected vector species it becomes obvious that some parts of the observed pathogen distribution cannot be explained when only *Phlebotomus* species with proven vector competence

(i.e. *P. argentipes*, *P. martini* and *P. orientalis*) are taken into account. *Phlebotomus celiae*, another proven vector species for *L. donovani*, which has not been considered in the modelling due to the lack of occurrence data, might fill in this gap. However, *P. celiae* occurs in savannah and forest areas in Kenya and Ethiopia [3] and is only known to act as a vector in Ethiopia [10]. *Phlebotomus vansomerenae* is also suspected to act as a vector for *L. donovani* in Kenya [10]. *Phlebotomus rodhaini*, on the other hand, is another suspected vector and was modelled to find suitable climatic conditions in large parts of the pathogen's distribution. Based on two female individuals of *P. rodhaini* infected with *L. donovani* and caught in eastern Sudan, Elnaiem *et al*. [20] suggested *P. rodhaini* being a possible vector of *L. donovani* for animal reservoir hosts but questioned its role in human infections. Among the considered 32 sandfly species, *P. rodhaini* is the only occurring vector species in some areas, and its role as an important vector seems evident. However, there may also be another disregarded vector-competent species that occurs in the region.

## 4.3. Limitations of the study approach

The purpose of the study was to compare the geographical distributions of *Phlebotomus* vector species and the distributions of their associated *Leishmania* pathogens and should lead to a better understanding of the vector–pathogen relationship, which could yield a better development of vector and disease surveillance and control programmes. However, the study is faced with some challenges, e.g. dealing with two different levels of distributional data: observed occurrence records (point data) for vector distribution and expert-based distribution areas (polygon data) for pathogen distribution. In order to bring this data into line, we have carried out an ecological niche modelling approach. In ENM approaches, some aspects should be kept in mind when interpreting the modelling results, but do not reduce the overall validity of the study. Models are only as good as the data they are based on. When using occurrence records as distributional data, one is usually confronted with a sampling bias that is also reflected in the modelling results. In some regions, the modelled vector distribution may be overestimated, as the species may be absent due to dispersal limitations (due to barriers like oceans or mountains) or due to relevant ecological factors disregarded in our approach (vegetation, soil, land cover, biotic interaction). The inclusion of vegetation indices (e.g. enhanced vegetation index (EVI)) and soil factors as predictor variables could have improved the modelling. The disregard of species' mobility in our approach could have been partly overcome by restricting the considered areas to those accessible for the respective sandfly species. But we decided in favour of a uniform approach for all species.

Besides these limitations, the model could result in an underestimation of the vector species' distribution in other regions for several reasons. These could be due to model overfitting or due to an underrepresentation of the species' distribution in the occurrence data (indicated e.g. by a low number of occurrence records or occurrence records being clustered in a certain region). For some species, the number of available occurrence records is so low (especially *P. langeroni*, *P. pedifer* and *P. salehli*), that it could reasonably be considered insufficient for the application of ENM. We decided to keep these species in our study in order for it to be as comprehensive as possible for the vector species.

To evaluate the ecological niche models, we used the AUC value. The AUC has the advantage of being a threshold-independent measure. However, in the case of sandflies, it could be argued that a measure that does not weight the two error components equally and penalizes omission error would be more appropriate (e.g. PartialROC as suggested by Peterson *et al*. [62]). A potential underestimation of the vector species distribution could be attributed to the choice of the threshold when transforming the continuous modelling result into binary maps, as the used threshold was found to yield the smallest area projected as suitable compared to other threshold selections [63]. In addition, the considered pathogen distribution is not beyond doubt, too. For example, *L. donovani* is known to be endemic in parts of India [64], but this area is not covered in our data.

Apart from the challenge that distributional data of different levels (point data versus polygon data) were compared, the available data refer to different time scales. Thus, the approach is based on the assumption that the spatial distribution patterns of the different components (vectors and pathogens) have not changed over time. This is a strong assumption, especially for species with low dispersal ability such as sandflies. Data availability remains a crucial limiting factor in correlative modelling approaches.

## 5. Conclusion

Distribution areas of pathogens that do not overlap with areas modelled to be suitable for vector species might hint towards incomplete entomological fieldwork of vector reporting, e.g. in remotes areas. Thus,

further monitoring the current distribution of sandfly species is of great importance and the joint actions of public health and veterinarian authorities in the sense of a 'One health approach' should be strengthened. Ecological niche modelling accounting for interactions among hosts, vectors and parasites offer powerful approaches for identifying regions potentially at risk for infection. Our approach can support risk assessments of leishmaniasis and thereby potentially help contain the disease. However, much more detail and consideration of other risk factors (e.g. human socioeconomic variables), are required for full development of risk maps [14]. If data become available, future analyses should also account these factors as well as for disease cases to further narrow down areas of risk.

Data accessibility. All relevant data have been uploaded as part of the supplementary material.

Authors' contributions. S.C. conceived of the study, carried out the analysis, created figures and wrote the manuscript. J.K. participated in the design of the study and wrote the manuscript. L.K.K. and K.J.Q.H. collected distributional data, participated in data analysis and creating figures and wrote the manuscript. S.K. conceived of the study and wrote the manuscript. All authors gave final approval for publication and agree to be held accountable for the work performed therein.

Competing interests. We have no competing interests.

Funding. The present study was supported by the Uniscientia Stiftung. The present study is also a product of the Centre for Translational Biodiversity Genomics (LOEWE-TBG) as part of the 'LOEWE—Landes-Offensive zur Entwicklung Wissenschaftlich-ökonomischer Exzellenz' programme of Hesse's Ministry of Higher Education, Research, and the Arts.

Acknowledgements. The present study was supported by the Uniscientia Stiftung. The present study is also a product of the Centre for Translational Biodiversity Genomics (LOEWE-TBG) as part of the 'LOEWE—Landes-Offensive zur Entwicklung Wissenschaftlich-ökonomischer Exzellenz' programme of Hesse's Ministry of Higher Education, Research, and the Arts.

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
