## [Reviewer comments · Royal Society Open Science]

Review History

RSOS-190334.R0 (Original submission)

Review form: Reviewer 1

Is the manuscript scientifically sound in its present form?

Yes

Are the interpretations and conclusions justified by the results?

Yes

Is the language acceptable?

Yes

Is it clear how to access all supporting data?

Yes

Do you have any ethical concerns with this paper?

No

Have you any concerns about statistical analyses in this paper?

No

Recommendation?

Accept with minor revision (please list in comments)

Comments to the Author(s)

I previously reviewed this same manuscript to another Journal. Again, I still see that the manuscript by Cunze et al. presents an interesting work on leishmaniasis in Eurasia and Africa. The authors clearly stated the objectives of their study to compare the geographical distributions of *Phlebotomus* vector species and their corresponding pathogens. The manuscript is in the line with these objectives and therefore represents a valuable contribution in the field of vector and distributional ecology. So, I see that the manuscript is worth to be published; many improvements were considered to the original version I reviewed previously for another Journal. Finally, my decision remains the same to my previous one and I strongly recommend the manuscript for publication in your Journal after considering some concerns that I will detail below:

- Line 31-34: The vector incrimination requires other factors to be considered before drawing a similar conclusion. So, I would kindly ask the author to change this piece to reflect only their finding based on a single factor of vector incrimination (co-occurrence/co-existence of both vector and pathogen in question).
- Line 63: Add "as one of the vector incrimination criteria adopted by WHO [3]".
- Line 75: This assumption is incorrectly stated; as I mentioned previously don't get yourself into such debates in this era. I would suggest the authors to use the previous literature to reflect that one of the reasons to exclude reservoirs in their analyses is the much broader ranges for these hosts (ref. 15 in your list "<https://doi.org/10.1371/journal.pntd.0004381>" and <https://doi.org/10.1016/j.actatropica.2018.08.014>).
- Line 82: Change "Lybia" to "Libya".
- Line 106-113: Cite supplementary materials in the text, particularly data used for modeling *Phlebotomus* spp.
- Line 116: Change "5 arc minutes" to "30 arc-seconds".
- Line 244: AUC is longer a sensitive tool to evaluate model robustness (see Lobo et al. 2008, and Peterson et al. 2008). Authors should consider a more robust tool to evaluate their models; you may use PartialROC (Peterson et al. 2008) or an independent set of records if available. I am aware that you placed great efforts on your analyses, so, I suggest to add a paragraph that place all study limitations together in the discussion and include this one as one of these limitations.
- Line 115 – 127: The environmental data used for the niche modeling analyses were limited and missing important variables, for example, EVI, and soil factors that are available for present-day conditions which are the main interest of this study. Thanks to the authors to include the statement on line 128-129; however, these factors are still important to estimate the ecological niches of these species. Please outline this one in the limitation section I previously referred (see my previous comment).
- Line 130-134: Several improvements have been introduced to the field of ecological modeling. These improvements solved some problems associated with species distribution modeling, particularly in the case of Maxent. None of these were considered in this analysis; estimating accessible area for model calibration of the species in question (See Barve et al. 2011 and Owens et al. 2013); some of these species have limited distribution to Europe, so, calibrating these models should be limited to areas accessible for the species (M component of the BAM diagram). Please consider adding this item to the limitations section.
- The limitations paragraph on line 338-348 marks an improvement for this manuscript version, so, I would suggest adding all missed items that I referred previously in my review.

Review form: Reviewer 2

Is the manuscript scientifically sound in its present form?

No

Are the interpretations and conclusions justified by the results?

No

Is the language acceptable?

Yes

Is it clear how to access all supporting data?

Yes

Do you have any ethical concerns with this paper?

No

Have you any concerns about statistical analyses in this paper?

I do not feel qualified to assess the statistics

Recommendation?

Major revision is needed (please make suggestions in comments)

Comments to the Author(s)

Dear Authors

Thanks for good idea for this study. It is recommended to see the comments put on the attached file (Appendix A) and revise this manuscript.

Decision letter (RSOS-190334.R0)

12-Apr-2019

Dear Dr Cunze

On behalf of the Editors, I am pleased to inform you that your Manuscript RSOS-190334 entitled "Leishmaniasis in Eurasia and Africa – geographical distribution of vector species and pathogens" has been accepted for publication in Royal Society Open Science subject to minor revision in accordance with the referee suggestions. Please find the referees' comments at the end of this email.

The reviewers and handling editors have recommended publication, but also suggest some minor revisions to your manuscript. Therefore, I invite you to respond to the comments and revise your manuscript.

- Ethics statement

- Data accessibility

If you wish to submit your supporting data or code to Dryad (<http://datadryad.org/>), or modify your current submission to dryad, please use the following link:
<http://datadryad.org/submit?journalID=RSOS&manu=RSOS-190334>

- Competing interests

- Authors' contributions

- Acknowledgements

- Funding statement

Because the schedule for publication is very tight, it is a condition of publication that you submit the revised version of your manuscript before 21-Apr-2019. Please note that the revision deadline

will expire at 00.00am on this date. If you do not think you will be able to meet this date please let me know immediately.

If your manuscript is newly submitted and subsequently accepted for publication, you will be asked to pay the article processing charge, unless you request a waiver and this is approved by

Royal Society Publishing. You can find out more about the charges at <http://rsos.royalsocietypublishing.org/page/charges>. Should you have any queries, please contact openscience@royalsociety.org.

on behalf of Dr Denise Greig (Associate Editor) and Kevin Padian (Subject Editor)
openscience@royalsociety.org

Associate Editor Comments to Author (Dr Denise Greig):

Associate Editor: 1

Comments to the Author:

This paper uses GIS to model the relationships between Leishmania parasite species and their sandfly vector species and to identify any mismatches based on known and expected (based on climate) occurrence data of parasite/vector pairs. This manuscript provides a thoughtful synthesis, discussion, and visualization of some complex data and the map figures are really well done.

I am not familiar with the Maxent software and defer to the two reviewers who had several insightful comments. Both pointed out limitations and recent improvements to Maxent, which, if you are not able to address through your analysis, should be described in the text so that readers know of those limitations.

Below are a few minor comments and suggestions on word choice:

Line 90. delete comma after the word "areas"

Line 126. First mention of Maxent here, but it is not described until line 130. Please correct.

Line 168. Replace "few" with "limited"

Line 206. I'm not sure that "exemplary" is the word you are looking for here. Maybe re-write this first sentence and say why you focused specifically on *L. donovani* and its associated vector species?

Line 283. I would insert "a" before species and add an "s" to the word "area" in the next line.

Line 291. Replace "and due to" with "or"

Line 304. I don't think "monotonously" is the word you mean here. Maybe "...the niches of the second group, including *L. donovani*, show optima at warmer temperatures"

Line 310. Change to "covers and extends beyond the known pathogen distribution." And delete "and goes further beyond"

Line 357. Maybe change "inappropriate" to "incomplete"?

Reviewer comments to Author:

Reviewer: 1

Comments to the Author(s)

I previously reviewed this same manuscript to another Journal. Again, I still see that the manuscript by Cunze et al. presents an interesting work on leishmaniasis in Eurasia and Africa. The authors clearly stated the objectives of their study to compare the geographical distributions

of *Phlebotomus* vector species and their corresponding pathogens. The manuscript is in the line with these objectives and therefore represents a valuable contribution in the field of vector and distributional ecology. So, I see that the manuscript is worth to be published; many improvements were considered to the original version I reviewed previously for another Journal. Finally, my decision remains the same to my previous one and I strongly recommend the manuscript for publication in your Journal after considering some concerns that I will detail below:

- Line 31-34: The vector incrimination requires other factors to be considered before drawing a similar conclusion. So, I would kindly ask the author to change this piece to reflect only their finding based on a single factor of vector incrimination (co-occurrence/co-existence of both vector and pathogen in question).
- Line 63: Add "as one of the vector incrimination criteria adopted by WHO [3]"
- Line 75: This assumption is incorrectly stated; as I mentioned previously don't get yourself into such debates in this era. I would suggest the authors to use the previous literature to reflect that one of the reasons to exclude reservoirs in their analyses is the much broader ranges for these hosts (ref. 15 in your list "<https://doi.org/10.1371/journal.pntd.0004381>" and <https://doi.org/10.1016/j.actatropica.2018.08.014>).
- Line 82: Change "Lybia" to "Libya".
- Line 106-113: Cite supplementary materials in the text, particularly data used for modeling *Phlebotomus* spp.
- Line 116: Change "5 arc minutes" to "30 arc-seconds".
- Line 244: AUC is longer a sensitive tool to evaluate model robustness (see Lobo et al. 2008, and Peterson et al. 2008). Authors should consider a more robust tool to evaluate their models; you may use PartialROC (Peterson et al. 2008) or an independent set of records if available. I am aware that you placed great efforts on your analyses, so, I suggest to add a paragraph that place all study limitations together in the discussion and include this one as one of these limitations.
- Line 115–127: The environmental data used for the niche modeling analyses were limited and missing important variables, for example, EVI, and soil factors that are available for present-day conditions which are the main interest of this study. Thanks to the authors to include the statement on line 128-129; however, these factors are still important to estimate the ecological niches of these species. Please outline this one in the limitation section I previously referred (see my previous comment).
- Line 130-134: Several improvements have been introduced to the field of ecological modeling. These improvements solved some problems associated with species distribution modeling, particularly in the case of Maxent. None of these were considered in this analysis; estimating accessible area for model calibration of the species in question (See Barve et al. 2011 and Owens et al. 2013); some of these species have limited distribution to Europe, so, calibrating these models should be limited to areas accessible for the species (M component of the BAM diagram). Please consider adding this item to the limitations section.
- The limitations paragraph on line 338-348 marks an improvement for this manuscript version, so, I would suggest adding all missed items that I referred previously in my review.

Reviewer: 2

Comments to the Author(s)

Dear Authors

Thanks for good idea for this study. It is recommended to see the comments put on the attached file and revise this manuscript.

Author's Response to Decision Letter for (RSOS-190334.R0)

See Appendix B.

Decision letter (RSOS-190334.R1)

30-Apr-2019

Dear Dr Cunze,

I am pleased to inform you that your manuscript entitled "Leishmaniasis in Eurasia and Africa – geographical distribution of vector species and pathogens" is now accepted for publication in Royal Society Open Science.

on behalf of Dr Denise Greig (Associate Editor) and Kevin Padian (Subject Editor)
openscience@royalsociety.org

Associate Editor Comments to Author (Dr Denise Greig):
Associate Editor: 1
Comments to the Author:
(There are no comments.)

Reviewer comments to Author:

Appendix A**ROYAL SOCIETY
OPEN SCIENCE****Leishmaniasis in Eurasia and Africa – geographical
distribution of vector species and pathogens**

Journal:	Royal Society Open Science
Manuscript ID	RSOS-190334
Article Type:	Research
Date Submitted by the Author:	26-Feb-2019
Complete List of Authors:	Cunze, Sarah; Goethe University Frankfurt Institute of Ecology Evolution and Diversity; LOEWE Centre for Translational Biodiversity Genomics Kochmann, Judith; Goethe University Frankfurt Institute of Ecology Evolution and Diversity; Senckenberg Biodiversity and Climate Research Centre Koch, Lisa; Goethe University Frankfurt Institute of Ecology Evolution and Diversity Hasselmann, Korbinian; Goethe University Frankfurt Institute of Ecology Evolution and Diversity Klimpel, Sven; Goethe University Frankfurt Institute of Ecology Evolution and Diversity; LOEWE Centre for Translational Biodiversity Genomics; Senckenberg Biodiversity and Climate Research Centre
Subject:	health and disease and epidemiology < BIOLOGY
Keywords:	Phlebotomus , GIS overlay, Maxent, ecological niche modelling, neglected tropical diseases, sandflies
Subject Category:	Biology (whole organism)

**1 Leishmaniasis in Eurasia and Africa – geographical distribution of vector species and**
**2 pathogens**

4 Sarah Cunze*^{1,2}, Judith Kochmann^{1,3}, Lisa K. Koch¹, Korbinian J. Q. Hasselmann¹, Sven
5 Klimpel^{1,2,3}

¹Goethe-University, Institute of Ecology, Evolution and Diversity; Max-von-Laue-Str. 13, D-
60438 Frankfurt/ M.

²LOEWE Centre for Translational Biodiversity Genomics; Senckenberganlage 25, 60325
Frankfurt, Germany.

³Senckenberg Biodiversity and Climate Research Centre; Senckenberganlage 25, D-60325
Frankfurt/ M., Germany

sarahcunze@gmail.com

Judith.Kochmann@senckenberg.de

16 L.Koch@bio.uni-frankfurt.de

hasselmann.korbinian@web.de

Klimpel@bio.uni-frankfurt.de

*Corresponding author: Sarah Cunze, Goethe-University, Institute for Ecology, Evolution and
Diversity; Max-von-Laue-Str. 13, D-60438 Frankfurt/ M., Germany, sarahcunze@gmail.com

**23 Abstract**

Leishmaniasis is a vector-borne disease with a broad global occurrence and an increasing number
of recorded cases; however, it is still one of the world's most neglected diseases. We here provide
climatic suitability maps generated by means of an ecological niche modelling approach for 32
*Phlebotomus* vector species with proven or suspected vector competence for five *Leishmania*
pathogens occurring in Eurasia and Africa. A GIS-based spatial overlay analysis was then used to
compare the distributional patterns of vectors and pathogens to help evaluate the vector species-
pathogen-relationship currently found in the literature. Most of the pathogens occurred with at
least one of the associated vector species. In the case of *L. donovani*, only a not yet confirmed
vector species, *P. rodhaini*, could explain the occurrence of the pathogen in regions of Africa.
*Phlebotomus alexandri* and *Phlebotomus longiductus* on the other hand, proven vector species of
*L. donovani*, do not seem to qualify as vectors for the pathogen. Their distribution is restricted to
northern latitudes and does not match the pathogen's distribution, which lies in southern latitudes.
Other more locally confined mismatches were discussed for each pathogen species respectively.
The comparative geographical GIS-overlay of vector species and pathogens functions as a first
indication that testing and re-evaluation of some pathogen-vector relationships might be
worthwhile to improve risk assessments of leishmaniasis.

Introduction

Leishmaniasis is a vector-borne disease with a broad global distribution and an increasing
number of recorded cases worldwide [1,2]. However, it is still one of the world's most neglected
diseases [3]. Over the last decades the disease has been found to expand geographically with a
global increase of cases of visceral and cutaneous leishmaniasis [4] increasing the public health
problems associated with the disease epidemics. The reported range expansion of the diseases has
been associated with range expansions of vector populations in response to climate change [5].
Leishmaniasis is caused by protozoan parasites of the genus *Leishmania*. The transmission can
either be zoonotic and/or anthroponotic through the bite of an infected female phlebotomine
sandfly [6]. In Eurasia and Africa, all vector-competent sandfly species belong to the genus
*Phlebotomus* [7]. Cutaneous leishmaniasis (CL) is the most common form of leishmaniasis. In
the “old world” , it is caused by five currently recognized *Leishmania* species: *L. infantum*, *L.*
*tropica*, *L. major*, *L. aethiopica* and *L. donovani*. Visceral leishmaniasis (VL), another common
and more severe form of leishmaniasis, is only associated to the *Leishmania* species *L. infantum*
and *L. donovani* [8–11]. The specific *Leishmania* species cause different clinical symptoms in
humans [12,13].

A successful transmission requires presence of pathogen, vector, and host species [14]. Thus,
when estimating disease risk the occurrence of the vector species in addition to the distributions
of the disease-causing pathogen and the host species are of major importance (see [14,15]). For a
better understanding of the epidemiology of vector-borne diseases, comparisons between the
geographical distributions of vectors, pathogens, and disease cases using geographical
information systems (GIS) have been suggested as very valuable [3].

Another important component required for the completion of the *Leishmania* life cycle is the
reservoir host [15]. Generally, the pathogen-vector-reservoir host relations are rather complex
and have been shown to vary regionally and temporally. Several species, from small to large,
wild, domestic and synanthropic mammals have been reported as reservoir hosts for *Leishmania*
spp., among them rock hyraxes (especially for the Indian subcontinent with India, Nepal,
Bangladesh and east Africa with Ethiopia, Kenya), rodents, foxes, dogs, cats, and other domestic
animals [6]. Among the five *Leishmania* species considered here, *L. donovani* and *L. tropica* are
assumed to be anthroponotic with humans as reservoir hosts, having human-to-human
transmission through the vector, but there is some evidence for the possible involvement of
zoonotic transmission as well [6]. Although the presence or absence of reservoirs is crucial for
the distribution of leishmaniasis, we decided not to account for reservoir hosts in our study,
assuming that the presence of reservoir host species is unlikely to be a limiting factor.

[revised manuscript text omitted]
 (Fig. 3). This means that some parts of the observed distribution of *L. donovani* only match
with areas modelled to be climatically suitable for *P. rodhaini* but none of the other known
associated vector species. 51

The aforementioned differences between the two groups of vector species associated with *L.*
*donovani* in the distributional patterns in the geographical space are also reflected by differences
in the modelled temperature niche functions (Fig. 4). The modelled optima of annual mean
temperature (bio01) is lower for the northern group of vector species (*P. alexandri* and *P.*
*longiductus*), whereas curves for species found in more tropical climates with monotonously
increasing temperature niche curves are more similar to the optimum curve of the pathogen *L.*
*donovani* (Fig. 4). Overall, it can be noted that the southern group of vector species is closer
connected to the pathogen *L. donovani* regarding its geographical distribution as well as the
modelled temperature niche than the northern group of vector species.

Discussion

Leishmaniasis is an important neglected tropical disease [7] and was chosen here to compare
distributional patterns of pathogens and vectors. A GIS-based spatial overlay analysis was
applied, integrating observed distribution data of pathogens and modelled climatic suitability for
the vector species, to enable verification, at least to some degree, of the known or suspected
vector-pathogen relationships cited in the literature.

[revised manuscript text omitted]

All relevant data are provided in the supplementary material.

**Competing Interests**

We have no competing interests.

**Authors' Contributions**

SC conceived of the study, carried out the analysis, created figures and wrote the manuscript. JK

participated in the design of the study and wrote the manuscript. LKK collected distributional

data, participated in data analysis and creating figures and wrote the manuscript. KJQH collected

distributional data, participated in data analysis and creating figures and wrote the manuscript.

SK conceived of the study and wrote the manuscript. All authors gave final approval for

publication and agree to be held accountable for the work performed therein.

**Funding**

The present study was supported by the Uniscientia Stiftung. The present study is also a product

of the Centre for Translational Biodiversity Genomics (LOEWE-TBG) as part of the “LOEWE—

Landes-Offensive zur Entwicklung Wissenschaftlich-ökonomischer Exzellenz” programme of

Hesse’s Ministry of Higher Education, Research, and the Arts.

**Research Ethics**

We were not required to complete an ethical assessment prior to conducting our research.

**Animal Ethics**

We were not required to complete an ethical assessment prior to conducting our research.

**Permission to carry out fieldwork**

No permissions were required prior to conducting our research.

**Acknowledgements**

The present study was supported by the Uniscientia Stiftung. The present study is also a product

of the Centre for Translational Biodiversity Genomics (LOEWE-TBG) as part of the “LOEWE—

Landes-Offensive zur Entwicklung Wissenschaftlich-ökonomischer Exzellenz” programme of

Hesse’s Ministry of Higher Education, Research, and the Arts.

**References**

- [1] Pigott, D. M., Bhatt, S., Golding, N., Duda, K. A., Battle, K. E., Brady, O. J., Messina, J. P.,
Balard, Y., Bastien, P. & Pratlong, F. *et al.* 2014 Global distribution maps of the
leishmaniasis. *eLife* **3**.
- [2] Pigott, D. M., Golding, N., Messina, J. P., Battle, K. E., Duda, K. A., Balard, Y., Bastien, P.,
Pratlong, F., Brownstein, J. S. & Freifeld, C. C. *et al.* 2014 Global database of leishmaniasis
occurrence locations, 1960-2012. *Scientific data* **1**, 140036.
- [3] WHO. 2010 Control of the leishmaniasis. Report of a meeting of the WHO expert
committee on the control of leishmaniasis: World Health Organization.
- [4] Shaw, J. 2007 The leishmaniasis - survival and expansion in a changing world. A mini-
review. *Memórias do Instituto Oswaldo Cruz* **102**, 541–547.
- [5] Boussaa, S., Kahime, K., Samy, A. M., Salem, A. B. & Boumezzough, A. 2016 Species
composition of sand flies and bionomics of *Phlebotomus papatasi* and *P. sergenti* (Diptera:
Psychodidae) in cutaneous leishmaniasis endemic foci, Morocco. *Parasites & vectors* **9**, 60.
- [6] Alemayehu, B. & Alemayehu, M. 2017 Leishmaniasis. A Review on Parasite, Vector and
Reservoir Host. *Health Sci J* **11**.
- [7] Torres-Guerrero, E., Quintanilla-Cedillo, M. R., Ruiz-Esmenjaud, J. & Arenas, R. 2017
Leishmaniasis: a review. *FL1000Research* **6**, 750.
- [8] Chappuis, F., Sundar, S., Hailu, A., Ghalib, H., Rijal, S., Peeling, R. W., Alvar, J. &
Boelaert, M. 2007 Visceral leishmaniasis: what are the needs for diagnosis, treatment and
control? *Nature reviews. Microbiology* **5**, 873–882.
- [9] Alvar, J., Vélez, I. D., Bern, C., Herrero, M., Desjeux, P., Cano, J., Jannin, J. & den Boer, M.
2012 Leishmaniasis worldwide and global estimates of its incidence. *PLoS ONE* **7**, e35671.

[10] Maroli, M., Feliciangeli, M. D., Bichaud, L., Charrel, R. N. & Gradoni, L. 2013
Phlebotomine sandflies and the spreading of leishmaniasis and other diseases of public
health concern. *Medical and veterinary entomology* **27**, 123–147.
[11] Ready, P. D. 2014 Epidemiology of visceral leishmaniasis. *Clinical epidemiology* **6**, 147–
154.
[12] Aronson, N., Herwaldt, B. L., Libman, M., Pearson, R., Lopez-Velez, R., Weina, P.,
Carvalho, E., Ephros, M., Jeronimo, S. & Magill, A. 2017 Diagnosis and Treatment of
Leishmaniasis: Clinical Practice Guidelines by the Infectious Diseases Society of America
(IDSA) and the American Society of Tropical Medicine and Hygiene (ASTMH). *The*
*American journal of tropical medicine and hygiene* **96**, 24–45.
[13] Gradoni, L., López-Vélez, R. & Mokni, M. 2017 Manual on case management and
surveillance of the leishmaniasis in the WHO European region:
[http://www.euro.who.int/__data/assets/pdf_file/0006/341970/MANUAL-ON-CASE-](http://www.euro.who.int/__data/assets/pdf_file/0006/341970/MANUAL-ON-CASE-MANAGEMENT_FINAL_with-cover-and-ISBN.pdf?ua=1)
[MANAGEMENT_FINAL_with-cover-and-ISBN.pdf?ua=1](http://www.euro.who.int/__data/assets/pdf_file/0006/341970/MANUAL-ON-CASE-MANAGEMENT_FINAL_with-cover-and-ISBN.pdf?ua=1).
[14] Samy, A. M., Campbell, L. P. & Peterson, A. T. 2014 Leishmaniasis transmission:
distribution and coarse-resolution ecology of two vectors and two parasites in Egypt. *Revista*
*da Sociedade Brasileira de Medicina Tropical* **47**, 57–62.
[15] Samy, A. M., Annajar, B. B., Dokhan, M. R., Boussaa, S. & Peterson, A. T. 2016 Coarse-
resolution Ecology of Etiological Agent, Vector, and Reservoirs of Zoonotic Cutaneous
Leishmaniasis in Libya. *PLoS neglected tropical diseases* **10**, e0004381.
[16] WHO. 1989 *Geographical distribution of arthropod-borne diseases and their principal*
*vectors*. <http://www.ciesin.org/docs/001-613/001-613.html>. Accessed 8 May 2017.
[17] van den Enden, E. 2004 *Tropical medicine; illustrated lecture notes [CD-ROM]. Leishmania*
*distribution*. Antwerp: ITGPress.

[18] WHO. 2010 *Control of the Leishmaniases. Report of a Meeting of the WHO Expert*
*Committee on the Control of Leishmaniases*. WHO Technical Report Series. Geneva: World
Health Organization.
[19] Aoun, K. & Bouratbine, A. 2014 Cutaneous leishmaniasis in North Africa: a review.
*Parasite* **21**, 14.
[20] Artemiev, V. M. & Neronov, V. M. 1984 *Distribution and ecology of sandflies of the Old*
*World (genus Phlebotomus)*. Moscow: The USSR Committee for the Unesco Programme on
*Man and the Biosphere (MAB) Institute of Evolutionary Morphology and Animal Ecology;*
*USSR Academy of Science*.
[21] Benabdennbi, I., Pesson, B., Cadi-Soussi, M. & Marquez, F. M. 1999 Morphological and
isoenzymatic differentiation of sympatric populations of *Phlebotomus perniciosus* and
*Phlebotomus longicuspis* (Diptera Psychodidae) in Northern Morocco. *Journal of Medical*
*Entomology* **36**, 116–120.
[22] Weise, M. 2004 Reisetiermedizinisch und epidemiologisch wichtige Arten der kaninen
Parasitenfauna in europäischen Anrainerstaaten des Mittelmeeres und in Portugal für Hunde
in Deutschland. Dissertation. München.
[23] Steinhauser, I. 2005 Untersuchung zur Verbreitung von Sandmücken (Phlebotomen) in
Deutschland mit Hilfe geographischer Informationssysteme (GIS). Diploma thesis. Bonn.
[24] Boussaa, S., Boumezzough, A., Remy, P. E., Glasser, N. & Pesson, B. 2008 Morphological
and isoenzymatic differentiation of *Phlebotomus perniciosus* and *Phlebotomus longicuspis*
(Diptera Psychodidae) in Southern Morocco. *Acta tropica* **106**, 184–189.
[25] Rassi, Y., Javadian, E., Nadim, A., Rafizadeh, S., Zahraii, A., Azizi, K. & Mohebbali, M.
2009 *Phlebotomus perfiliewi transcaucasicus* a Vector of *Leishmania infantum* in
Northwestern Iran. *J Med Entomol* **46**, 1094–1098.

[26] Svobodová, M., Alten, B., Zídková, L., Dvořák, V., Hlavačková, J., Myšková, J., Šeblová,
478 V., Kasap, O. E., Belen, A. & Votýpka, J. 2009 Cutaneous leishmaniasis caused by
479 *Leishmania infantum* transmitted by *Phlebotomus tobbi*. *International journal for*
*parasitology* **39**, 251–256.
[27] Tarallo, V. D., Dantas-Torres, F., Lia, R. P. & Otranto, D. 2010 Phlebotomine sand fly
population dynamics in a leishmaniasis endemic peri-urban area in southern Italy. *Acta*
*tropica* **116**, 227–234.
[28] Tabbabi, A., Bousslimi, N., Rhim, A., Aoun, K. & Bouratbine, A. 2011 First report on
natural infection of *Phlebotomus sergenti* with *Leishmania* promastigotes in the cutaneous
leishmaniasis focus in southeastern Tunisia. *The American journal of tropical medicine and*
*hygiene* **85**, 646–647.
[29] Akhoundi, M., Parvizi, P., Baghaei, A. & Depaquit, J. 2012 The subgenus *Adlerius*
*Nitzulescu* (Diptera, Psychodidae, Phlebotomus) in Iran. *Acta tropica* **122**, 7–15.
[30] Berdjane-Brouk, Z., Charrel, R. N., Hamrioui, B. & Izri, A. 2012 First detection of
*Leishmania infantum* DNA in *Phlebotomus longicuspis* Nitzulescu, 1930 from visceral
leishmaniasis endemic focus in Algeria. *Parasitology research* **111**, 419–422.
[31] Giorgobiani, E., Lawyer, P. G., Babuadze, G., Dolidze, N., Jochim, R. C., Tskhvaradze, L.,
Kikaleishvili, K. & Kamhawi, S. 2012 Incrimination of *Phlebotomus kandelakii* and
*Phlebotomus balcanicus* as vectors of *Leishmania infantum* in Tbilisi, Georgia. *PLoS*
*neglected tropical diseases* **6**, e1609.
[32] Hijmans, R., Cameron, S., Parra, J., Jones, P. & Jarvis, A. WORLDCLIM - a set of global
climate layers (climate grids).

[33] Kasap, O. E. & Alten, B. 2005 Laboratory estimation of degree-day developmental
requirements of *Phlebotomus papatasi* (Diptera: Psychodidae). *Journal of vector ecology* :
*journal of the Society for Vector Ecology* **30**, 328–333.
[34] Kasap, O. E. & Alten, B. 2006 Comparative demography of the sand fly *Phlebotomus*
*papatasi* (Diptera: Psychodidae) at constant temperatures. *Journal of vector ecology* : *journal*
*of the Society for Vector Ecology* **31**, 378–385.
[35] Medlock, J. M., Hansford, K. M., van Bortel, W., Zeller, H. & Alten, B. 2014 A summary of
the evidence for the change in European distribution of phlebotomine sand flies (Diptera:
Psychodidae) of public health importance. *Journal of vector ecology* : *journal of the Society*
*for Vector Ecology* **39**, 72–77.
[36] Elith, J., Phillips, S. J., Hastie, T., Dudík, M., Chee, Y. E. & Yates, C. J. 2011 A statistical
explanation of MaxEnt for ecologists. *Diversity and Distributions* **17**, 43–57.
[37] Phillips, S. J., Dudík, M. & Schapire, R. E. *Maxent software for modeling species niches and*
*distributions*.
[38] Cunze, S. & Tackenberg, O. 2015 Decomposition of the maximum entropy niche function –
A step beyond modelling species distribution. *Environmental Modelling & Software* **72**,
250–260.
[39] Liu, C., White, M., Newell, G. & Pearson, R. 2013 Selecting thresholds for the prediction of
species occurrence with presence-only data. *Journal of Biogeography* **40**, 778–789.
[40] Jiménez-Valverde, A. & Lobo, J. M. 2007 Threshold criteria for conversion of probability of
species presence to either–or presence–absence. *Acta Oecologica* **31**, 361–369.
[41] Lobo, J. M., Jiménez-Valverde, A. & Real, R. 2008 AUC. A misleading measure of the
performance of predictive distribution models. *Global Ecol Biogeography* **17**, 145–151.

[42] R Core Team. 2017 *R: A language and environment for statistical computing*. Vienna,
Austria: R Foundation for Statistical Computing.
[43] *ArcGIS*: ESRI.
[44] Melaun, C., Kruger, A., Werblow, A. & Klimpel, S. 2014 New record of the suspected
leishmaniasis vector *Phlebotomus (Transphlebotomus) mascittii* Grassi, 1908 (Diptera:
Psychodidae: Phlebotominae)--the northernmost phlebotomine sandfly occurrence in the
Palearctic region. *Parasitology research* **113**, 2295–2301.
[45] Naucke, T. J. 2002 Leishmaniose, eine Tropenkrankheit und deren Vektoren (Diptera,
Psychodidae, Phlebotominae) in Mitteleuropa. *Densia* **6**, 163–178.
[46] Berdjane-Brouk, Z., Charrel, R. N., Bitam, I., Hamrioui, B. & Izri, A. 2011 Record of
*Phlebotomus (Transphlebotomus) mascittii* Grassi, 1908 and *Phlebotomus (Larroussius)*
*chadlii* Rioux, Juminer & Gibily, 1966 female in Algeria. *Parasite* **18**, 337–339.
[47] Obwaller, A. G., Karakus, M., Poepl, W., Toz, S., Ozbel, Y., Aspöck, H. & Walochnik, J.
2016 Could *Phlebotomus mascittii* play a role as a natural vector for *Leishmania infantum*?
New data. *Parasites & vectors* **9**, 458.
[48] Elnaiem, D.-E. A., Hassan, H. K., Osman, O. F., Maingon, R. D., Killick-Kendrick, R. &
Ward, R. D. 2011 A possible role for *Phlebotomus (Anaphlebotomus) rodhaini* (Parrot,
1930) in transmission of *Leishmania donovani*. *Parasites & vectors* **4**, 238.
[49] Aransay, A. M., Scoulica, E. & Tselentis, Y. 2000 Detection and identification of
*Leishmania* DNA within naturally infected sand flies by seminested PCR on Minicircle
Kinetoplastic DNA. *Applied and Environmental Microbiology* **66**, 1933–1938.
[50] Depaquit, J., Ferté, H., Léger, N., Lefranc, F., Alves-Pires, C., Hanafi, H., Maroli, M.,
Morillas-Marquez, F., Rioux, J.-A. & Svobodova, M. *et al.* 2002 ITS 2 sequences
heterogeneity in *Phlebotomus sergenti* and *Phlebotomus similis* (Diptera, Psychodidae).

Possible consequences in their ability to transmit *Leishmania tropica*. *International journal*
*for parasitology* **32**, 1123–1131.
[51] Fischer, D., Thomas, S. M. & Beierkuhnlein, C. 2011 Modelling climatic suitability and
dispersal for disease vectors. The example of a phlebotomine sandfly in Europe. *Procedia*
*Environmental Sciences* **7**, 164–169.
[52] Trájer, A. J. 2013 The effect of climate change on the potential distribution of the European
Phlebotomus species. *Applied Ecology and Environmental Research* **11**, 189–208.
[53] Chalhaf, B., Chlif, S., Mayala, B., Ghawar, W., Bettaieb, J., Harrabi, M., Benie, G. B.,
Michael, E. & Salah, A. B. 2016 Ecological Niche Modeling for the Prediction of the
Geographic Distribution of Cutaneous Leishmaniasis in Tunisia. *The American journal of*
555 *tropical medicine and hygiene* **94**, 844–851.
[54] Koch, L. K., Kochmann, J., Klimpel, S. & Cunze, S. 2017 Modeling the climatic suitability
of leishmaniasis vector species in Europe. *Scientific reports* **7**, 13325.
[55] Chalhaf, B., Chemkhi, J., Mayala, B., Harrabi, M., Benie, G. B., Michael, E. & Ben Salah,
35 A. 2018 Ecological niche modeling predicting the potential distribution of *Leishmania*
vectors in the Mediterranean basin: impact of climate change. *Parasites & vectors* **11**, 461.
[56] Yaghoobi-Ershadi. 2012 Phlebotomine Sand Flies (Diptera: Psychodidae) in Iran and their
Role on *Leishmania* Transmission. *Journal of arthropod-borne diseases* **6**, 1–17.
[57] Krayter, L., Schnur, L. F., Schönián, G. & Wang, T. 2015 The Genetic Relationship between
*Leishmania aethiopica* and *Leishmania tropica* Revealed by Comparing Microsatellite
Profiles. *PLoS ONE* **10**, e0131227.
[58] Anjili, C. O., Ngumbi, P. M., Kaburi, J. C. & Irungu, L. W. 2011 The phlebotomine sandfly
fauna (Diptera: Psychodidae) of Kenya. *Journal of vector borne diseases* **48**, 183–189.

[59] Fischer, D., Thomas, S. M. & Beierkuhnlein, C. 2010 Temperature-derived potential for the
establishment of phlebotomine sandflies and visceral leishmaniasis in Germany. *Geospatial*
*Health* **5**, 59–69.

[60] Norris, D. 2014 Model Thresholds are More Important than Presence Location Type.
Understanding the Distribution of Lowland tapir (*Tapirus Terrestris*) in a Continuous
Atlantic Forest of Southeast Brazil. *Tropical Conservation Science* **7**, 529–547.

— proven
 - - - suspected

Climatic suitability of vector species and distribution of associated pathogens. Hatched areas indicate the observed distribution of the five *Leishmania* species (according to [16–19]), red indicates areas with suitable climatic conditions projected for at least one of the associated vector species (with confirmed or strongly suspected vector-competence for the respective *Leishmania* species). See Fig. S1 for additional information on individual vector species occurrences.

285x146mm (300 x 300 DPI)

Geographical distribution of *Leishmania donovani* and its associated vector species. Hatched areas indicate
the observed distribution of *L. donovani*. Coloured areas indicate modelled suitability for at least one vector
species of *L. donovani*: orange = northern group (*Phlebotomus alexandri*, *Phlebotomus longiductus*), light
red = southern group (*Phlebotomus argentipes*, *Phlebotomus martini*, *Phlebotomus orientalis* and
*Phlebotomus rodhaini*), dark red = *Phlebotomus rodhaini* and no one of the other associated vector species.

293x182mm (300 x 300 DPI)

One-variable response curves (niche functions) for *Leishmania donovani* and associated vector species. Modelled climatic suitability depending on the annual mean temperature [°C] for the vector species (proven vector-competence – solid line, suspected vector-competence – dotted line) and the associated *Leishmania* pathogen (bold line).

225x171mm (300 x 300 DPI)

Appendix B

Dear Dr. Greig,

Thank you for your kind reply and the opportunity to review the manuscript. We incorporated and addressed the changes that you and the two reviewers made. We hope that the text was improved to your satisfaction.

Please find our responses in italics and indented below.

Kind regards,

Sarah Cunze, on behalf of all authors

Associate Editor Comments to Author (Dr Denise Greig): Associate Editor: 1

Comments to the Author:

This paper uses GIS to model the relationships between Leishmania parasite species and their sandfly vector species and to identify any mismatches based on known and expected (based on climate) occurrence data of parasite/vector pairs. This manuscript provides a thoughtful synthesis, discussion, and visualization of some complex data and the map figures are really well done.

I am not familiar with the Maxent software and defer to the two reviewers who had several insightful comments. Both pointed out limitations and recent improvements to Maxent, which, if you are not able to address through your analysis, should be described in the text so that readers know of those limitations.

Below are a few minor comments and suggestions on word choice:

Line 90. delete comma after the word “areas”

Done

Line 126. First mention of Maxent here, but it is not described until line 130. Please correct.

Done

Line 168. Replace “few” with “limited”

Done

Line 206. I’m not sure that “exemplary” is the word you are looking for here. Maybe re-write this first sentence and say why you focused specifically on *L. donovani* and its associated vector species?

Done

Line 283. I would insert “a” before species and add an “s” to the word “area” in the next line.

Done

Line 291. Replace “and due to” with “or”

Done

Line 304. I don’t think “monotonously” is the word you mean here. Maybe “...the niches of the second group, including *L. donovani*, show optima at warmer temperatures”

Done

Line 310. Change to “covers and extends beyond the known pathogen distribution.” And delete “and goes further beyond”

Done

Line 357. Maybe change “inappropriate” to “incomplete”?

Done

Reviewer comments to Author: Reviewer: 1

Comments to the Author(s)

I previously reviewed this same manuscript to another Journal. Again, I still see that the manuscript by Cunze et al. presents an interesting work on leishmaniasis in Eurasia and Africa. The authors clearly stated the objectives of their study to compare the geographical distributions of Phlebotomus vector species and their corresponding pathogens. The manuscript is in the line with these objectives and therefore represents a valuable contribution in the field of vector and distributional ecology. So, I see that the manuscript is worth to be published; many improvements were considered to the original version I reviewed previously for another Journal. Finally, my decision remains the same to my previous one and I strongly recommend the manuscript for publication in your Journal after considering some concerns that I will detail below:

We are grateful to Reviewer 1 for the kind and valuable comments and suggestions.

- Line 31-34: The vector incrimination requires other factors to be considered before drawing a similar conclusion. So, I would kindly ask the author to change this piece to reflect only their finding based on a single factor of vector incrimination (co-occurrence/co-existence of both vector and pathogen in question).

We placed a similar phrasing before the statement, but kept lines 31-34 unchanged. It now reads: "Based on this single factor of vector incrimination, that is, co-occurrence of both vector and pathogen, most of the pathogens occurred with at least one of the associated vector species. In the case of L. donovani"

- Line 63: Add "as one of the vector incrimination criteria adopted by WHO [3]".

Done

- Line 75: This assumption is incorrectly stated; as I mentioned previously don't get yourself into such debates in this era. I would suggest the authors to use the previous literature to reflect that one of the reasons to exclude reservoirs in their analyses is the much broader ranges for these hosts (ref. 15 in your list "<https://doi.org/10.1371/journal.pntd.0004381>" and <https://doi.org/10.1016/j.actatropica.2018.08.014>).

The sentence was changed and the references added. It now reads: "Although the presence or absence of reservoirs is crucial for the distribution of leishmaniasis, reservoir hosts were not included in the analysis. One of the reasons is their much broader range of distribution [15,16]."

- Line 82: Change "Lybia" to "Libya".

Done

- Line 106-113: Cite supplementary materials in the text, particularly data used for modeling Phlebotomus spp.

Done

- Line 116: Change "5 arc minutes" to "30 arc-seconds".

5 arc minutes is correct, but we changed 1 km into 10 km accordingly.

- Line 244: AUC is longer a sensitive tool to evaluate model robustness (see Lobo et al. 2008, and Peterson et al. 2008). Authors should consider a more robust tool to evaluate their

models; you may use PartialROC (Peterson et al. 2008) or an independent set of records if available. I am aware that you placed great efforts on your analyses, so, I suggest to add a paragraph that place all study limitations together in the discussion and include this one as one of these limitations.

As suggested by the reviewer we inserted a paragraph in the discussion that summarizes all limitations of the study addressing the PartialROC.

- Line 115—127: The environmental data used for the niche modeling analyses were limited and missing important variables, for example, EVI, and soil factors that are available for present-day conditions which are the main interest of this study. Thanks to the authors to include the statement on line 128-129; however, these factors are still important to estimate the ecological niches of these species. Please outline this one in the limitation section I previously referred (see my previous comment).

We have extended the last paragraph in the discussion and made it a separate section entitled “Limitations of the study approach”.

- Line 130-134: Several improvements have been introduced to the field of ecological modeling. These improvements solved some problems associated with species distribution modeling, particularly in the case of Maxent. None of these were considered in this analysis; estimating accessible area for model calibration of the species in question (See Barve et al. 2011 and Owens et al. 2013); some of these species have limited distribution to Europe, so, calibrating these models should be limited to areas accessible for the species (M component of the BAM diagram). Please consider adding this item to the limitations section.

Included in the section “Limitations of the study approach”

- The limitations paragraph on line 338-348 marks an improvement for this manuscript version, so, I would suggest adding all missed items that I referred previously in my review.

Done.

Reviewer: 2

Comments to the Author(s)

Dear Authors

Thanks for good idea for this study. It is recommended to see the comments put on the attached file and revise this manuscript.

We are grateful to Reviewer 2 for the kind and valuable comments and suggestions.

Comments have been transferred from the PDF into the response letter.

L101: Did you model the distribution of pathogens as well?

Clarified in the text (No we didn't.)

L 132: Please see two relevant references about the predictive bioclimatic variables on CL and VL vectors: (Predicted Distribution of Visceral Leishmaniasis Vectors (Diptera: Psychodidae; Phlebotominae) in Iran: A Niche Model Study. *Zoonoses and Public Health*, 2015, 62, 644–654

Modeling the Distribution of Cutaneous Leishmaniasis Vectors (Psychodidae: Phlebotominae) in Iran: A Potential Transmission in Disease Prone Areas. *J. Med. Entomol.* 52(4): 557–565 (2015).

We refer to these studies in the revised version.

L. 150: Do you mean you used the output of MaxEnt model as the layer of Leishmaniasis vector? If yes, please explain more about the threshold used for presence of sandfly vector? You know the output of model predicts between 0 and 1.

Clarified. (Yes; we considered areas where at least one vector is modelled to be present in a binary Maxent model)

167ff: In this section please just present the findings of this study. All discussions and interpretations should be moved to the Discussion section.

Done

177: Six species modelled in this study have less than 20 presence records:

P. chadi=12; P. langeroni=6; P. longipes=14; P. pedifer=6; P. salehi=8; P. similis=19. It seems more presence records are necessary for Maxent.

We discuss this item in the section "Limitations of the study approach"

203 The map of L. donovani missed India!!

Included in the section "Limitations of the study approach"

211 The maps of Leishmania parasites show an overview of the areas where the disease can be occurred. For example L. major foci didn't report from the east of Turkey and Northwest of Iran, but the map covered these areas. These maps are not 100 percent reliable. Please note this.

Done

Here is results, you should just present your findings. Please move all discussions and interpretations of your findings to the discussion section of this manuscript.

Done

This part is not discussion. It presented in Introduction and so it is suggested to remove it.

Done

There are more studies on sandflies using MaxEnt in the area you studied in this research. Please do a comprehensive review of literature and use the relevant references in your discussion.

We refer to the references you mentioned.

For modeling of sandflies, did you use all occurrence points for training the model? Which portion of point used as test data?

We decided not to split the data into trainings and test data sets because of the low numbers of available occurrence records for some species.

Please note you used few papers as resources of occurrence data for sandflies. So, you missed many records of sandflies in your study. This point also should be noted in discussion.

Included in the section "Limitations of the study approach"

Fig2: What does mean "modelled climatic suitability" here? Do you mean in Red areas, the environment is 100 percent suitable for at least one sandfly vector?

No, it doesn't. Area, suitable for at least one of the associated vector species according to the binary modelling results are displayed in red.

Clarified in the figure legend.

Fig3: I think you missed Indian continent as the endemic area for *L. donovani*. Please do more review of literature and revise this map.

We now discuss this in the section "Limitations of the study approach"